# Exploring The Prognostic Significance of SET-Domain Containing 2 (SETD2) Expression in Advanced and Castrate-Resistant Prostate Cancer

**DOI:** 10.3390/cancers16071436

**Published:** 2024-04-08

**Authors:** Yaser Gamallat, Joema Felipe Lima, Sima Seyedi, Qiaowang Li, Jon George Rokne, Reda Alhajj, Sunita Ghosh, Tarek A. Bismar

**Affiliations:** 1Department of Pathology and Laboratory Medicine, Cumming School of Medicine, University of Calgary, Calgary, AB T2N 4N1, Canada; yaser.gamallat@ucalgary.ca (Y.G.); joema.felipelima@ucalgary.ca (J.F.L.); sima.seyedi@ucalgary.ca (S.S.); 2Departments of Oncology, Biochemistry and Molecular Biology, Cumming School of Medicine, University of Calgary, Calgary, AB T2N 4N1, Canada; 3Arnie Charbonneau Cancer Institute, Cumming School of Medicine, University of Calgary, Calgary, AB T2N 4N1, Canada; 4Department of Computer Science, University of Calgary, Calgary, AB T2N 1N4, Canada; qiaowang.li@ucalgary.ca (Q.L.);; 5Department of Computer Engineering, Istanbul Medipol University, 34810 Istanbul, Turkey; 6Department of Health Informatics, University of Southern Denmark, 5230 Odense, Denmark; 7Department of Medical Oncology, College of Health Sciences, University of Alberta, Edmonton, AB T6G 2R7, Canada; 8Department of Public Health Sciences, Henry Ford Health, Detroit, MI 48202, USA; 9Tom Baker Cancer Center, Alberta Health Services, Calgary, AB T2N 4N1, Canada; 10Prostate Cancer Centre, Alberta Health Services, Calgary, AB T2V 1P9, Canada; 11Department of Pathology, Alberta Precision Labs, Calgary, AB T2V 1P9, Canada

**Keywords:** SETD2, prostate cancer, overall survival, cause specific survival, ERG, PTEN

## Abstract

**Simple Summary:**

SETD2, a histone methyltransferase and epigenetic modifier, and SETD2 protein expression were explored in a prostate cancer non-surgical cohort of 202 cases. Notably, SETD2 showed higher intensity in advanced and castrate-resistant disease compared to incidental cases. Moreover, elevated SETD2 expression is significantly associated with poorer prognosis, lower overall survival (OS), and decreased cancer-specific survival (CSS). High-risk SETD2 combined with PTEN loss or ERG positivity improved the prognostication for these outcomes. Additionally, the TCPA protein database and TCGA PRAD GSEA implicated SETD2 in pathways linked to tumor progression, chemoresistance, and adverse prognosis, including the AMPK, cAMP, and PI3K-Akt signaling pathways.

**Abstract:**

SET-domain containing 2 (SETD2) is a histone methyltransferase and an epigenetic modifier with oncogenic functionality. In the current study, we investigated the potential prognostic role of SETD2 in prostate cancer. A cohort of 202 patients’ samples was assembled on tissue microarrays (TMAs) containing incidental, advanced, and castrate-resistant CRPCa cases. Our data showed significant elevated SETD2 expression in advanced and castrate-resistant disease (CRPCa) compared to incidental cases (2.53 ± 0.58 and 2.21 ± 0.63 vs. 1.9 ± 0.68; *p* < 0.001, respectively). Interestingly, the mean intensity of SETD2 expression in deceased vs. alive patients was also significantly different (2.31 ± 0.66 vs. 2 ± 0.68; *p* = 0.003, respectively). Overall, high SETD2 expression was found to be considered high risk and was significantly associated with poor prognosis and worse overall survival (OS) (HR 1.80; 95% CI: 1.28–2.53, *p* = 0.001) and lower cause specific survival (CSS) (HR 3.14; 95% CI: 1.94–5.08, *p* < 0.0001). Moreover, combining high-intensity SETD2 with PTEN loss resulted in lower OS (HR 2.12; 95% CI: 1.22–3.69, *p* = 0.008) and unfavorable CSS (HR 3.74; 95% CI: 1.67–8.34, *p* = 0.001). Additionally, high SETD2 intensity with ERG positive expression showed worse prognosis for both OS (HR 1.99, 95% CI 0.87–4.59; *p* = 0.015) and CSS (HR 2.14, 95% CI 0.98–4.68, *p* = 0.058). We also investigated the protein expression database TCPA, and our results showed that high SETD2 expression is associated with a poor prognosis. Finally, we performed TCGA PRAD gene set enrichment analysis (GSEA) data for SETD2 overexpression, and our data revealed a potential association with pathways involved in tumor progression such as the AMPK signaling pathway, the cAMP signaling pathway, and the PI3K-Akt signaling pathway, which are potentially associated with tumor progression, chemoresistance, and a poor prognosis.

## 1. Introduction

Prostate cancer (PCa) is one of the most common cancers among men, with an estimated 288,300 new cases in 2023 in the USA alone. Since 2014, the incidence rate has increased by 3% per year overall and by about 5% per year for advanced-stage prostate cancer [1]. Given the clinical need for potential diagnostic and prognostic tools to enhance therapeutic strategies and reduce the necessity for biopsies in risk prediction, biomarker discovery has been rapidly expanding in recent years. However, identification of high-risk prostate cancer (PCa) cases, especially those with a higher rate of progression and metastasis, and aid in better predicting tumor aggressiveness remain essential [2].

Few candidate biomarkers derived from genetic aberrations are clinically implicated in aggressive PCa behavior. Among these markers, the ETS-related gene (*ERG*), loss of phosphatase and tensin homolog (*PTEN*), and mutations of genes such as *SPOP*, *TP53,* and *RB1* are being extensively studied and evaluated as possible predictors of disease progression. Earlier, we established small tissue-based signatures such as ERG/PTEN or PTEN/ATM, which have been studied by our group as potential candidates for tissue-based prognostic signatures [3]. However, there is a persistent need to identify additional biomarkers that could potentially facilitate early diagnosis or predict therapeutic benefits.

Histone methyltransferase SET-domain containing 2 (*SETD2*), a gene located at chromosome 3 p21.31 [4], has been documented to be involved in epigenetic regulation, mainly by catalyzing histone methylation of H3K36me3 [5]. Several studies have reported its biological function as a tumor suppressor. *SETD2* has been attracting a lot of interest in recent years due to its involvement in tumor initiation and progression. The increased level of *SETD2* has been identified as crucial for the restoration of chromatin structure after transcription, thereby maintaining genomic integrity and stability [6,7].

*SETD2* is an epigenetic modifier with oncogenic functionality. Considering its significant role in epigenetic regulation, *SETD2* expression levels exhibit considerable heterogeneity across different tumor types, and its dysregulation has been associated with distinct prognostic outcomes such as increased tumor aggressiveness, metastatic potential, and resistance to therapy. Mutations of *SETD2* were observed and extensively studied in certain malignancies, such as clear-cell renal-cell carcinoma (ccRCC), colorectal cancer, lungs, gastric, glioma, and colorectal cancer [8,9,10,11]. However, SETD2 protein expression is not well explored in most of these studies. The TCGA data show a differential expression trend that underscores the complexity and tumor-dependent nature of *SETD2* dysregulation in cancer pathogenesis, highlighting the need for further research to elucidate its precise mechanisms and therapeutic implications across diverse tumor types [7,12,13].

The first mutations on the *SETD2* gene were reported back in 2010 by Dalgliesh et al. and were studied in clear-cell renal-cell carcinoma [14]. Before that, Sarakbi et al. found a negative association between SETD2 expression levels and advanced tumor stages in breast cancer [11]. High-grade gliomas in the brains of pediatric and young adult patients have also been studied [15]. In addition, whole-exome sequencing studies have unveiled somatic mutations in the *SETD2* gene across various malignant tumors; this suggests that *SETD2* inactivation is linked to tumorigenesis in these organs, even if less frequently [16].

Inactivation of *SETD2* has been implicated in the failure to localize damaged DNA by the repair machinery within the cell, as mentioned earlier. This deficiency in localization can result in genomic instability, which is recognized as one of the hallmarks of tumorigenesis [17].

Moreover, mutations in the *SETD2* gene or its loss of function can induce protein dysfunction, resulting in microsatellite instability and an increased frequency of spontaneous mutations. These events are implicated in tumorigenesis, progression to advanced prostate cancer, and metastasis [13,18].

In the current study, we have investigated the potential prognostic role of SETD2 in prostate cancer and its associations with other candidates’ biomarkers, such as PTEN, ERG, and p53.

## 2. Methodology

### 2.1. Tissue Microarray Construction

To investigate the role of SETD2 in prostate disease progression, 202 men were diagnosed with prostate cancer through transurethral resection of the prostate (TURP). Patients in this cohort either did not receive active treatment or were treated with androgen deprivation therapy (ADT) either before or after TURP. Those receiving treatments post-TURP were classified as the advanced group, while individuals with treatments prior to the TURP sample exhibiting advanced local disease with obstructive symptoms while on ADT were categorized as the CRPC group. The incidental group consisted of patients with no prior ADT therapy and who had Gleason grade groups 1–3.

The tissue microarray (TMA) was constructed consisting of incidental (*n* = 61, 30.2%), advanced (*n* = 69, 34.2%), and castrate-resistant prostate cancer (CRPC) (*n* = 71, 35.6%).

The demographics of patients, distribution according to the three subgroups, Gleason groups (GGs), and additional biomarker groups are described in Table 1. This study was approved by the Cumming School of Medicine Ethics Review Board, University of Calgary, in accordance with the 1964 Helsinki Declaration and its later amendments and comparable ethical standards. Clinical follow-up was obtained from the Alberta Tumor Registry and included dates of therapy, overall survival (OS), and prostate cancer-specific survival (CSS) and was approved by the University of Calgary, Cumming School of Medicine Ethics Review Board. Each sample diagnosis and GG were confirmed by pathologists in this study. Tissue samples from the cohort were assembled on two tissue microarrays (TMAs) with an average of two cores per patient using a manual tissue arrayer (Beecher Instruments, Silver Spring, MD, USA).

### 2.2. Immunohistochemistry

SETD2 protein expression was assessed on TMAs using immunohistochemistry (IHC) at the anatomical pathology lab of the Alberta Precision Research Lab (APRL) facility using a Dako Omnis autostainer. Briefly, about 4 µm formalin-fixed paraffin-embedded (FFPE) sections were deparaffinized and incubated with epitope retrieval buffer. Then, rabbit polyclonal SETD2 antibody (Cat # HPA042451, RRID: AB_10806239 Sigma-Aldrich, St. Louis, MO, USA) was used at a dilution of 1:50. TP53 expression was assessed using p53 antibody (DO-1): sc-126, Santa Cruz Biotechnology, Inc., Santa Cruz, CA, USA (1:50). ERG and PTEN were assessed using IHC. The FLEX DAB+ substrate chromogen system was used as a detection reagent (Agilent, Santa Clara, CA, USA). Counter-staining was performed using hematoxylin, followed by dehydration and mounting using Flo-TEXX mounting medium (Lerner Laboratories, Pittsburgh, PE, USA).

### 2.3. Pathological Assessment

The histological diagnosis of individual TMA cores was confirmed by two pathologists. Gleason scoring was evaluated according to the 2018 World Health Organization/International Society of Urological Pathology GGs. SETD2 protein expression was classified using a four-tiered system nuclear pattern (0 = negative, 1 = weak, 2 = moderate, and 3 = high expression). PTEN IHC was assessed using a four-tiered system: 0 = negative, 1 = weak, 2 = moderate, and 3 = high intensities. Data were grouped as binary based on previous risk group stratification (score 0 = high risk; complete absence of PTEN expression) and (scores 1, 2, and 3 = low risk; weak, moderate, and high PTEN intensity, respectively). ERG IHC was evaluated as binary values (negative vs. positive) reflective of the presence or absence of ERG gene rearrangements. We assessed TP53 status using a previously validated method [19,20], which reflected TP53 sequencing mutations: score 1 = wild type, nuclear staining (strong or weak) with internal control; score 0 = absent nuclear staining with positive control; score 2 = overexpressed nuclear staining; and score 3 = cytoplasmic staining.

### 2.4. Bioinformatics and Public Database Analysis

Prostate Cancer proteomics utilized TCPA Portal protein-level 4 data from the reverse-phase protein arrays (RPPA) containing prostate cancer (*n* = 351 samples) [21]. Data analysis performed using webtool TRGAted (accessed on 12 December 2023) [22] was scaled to *z*-scores and overall survival (OS), and disease-specific survival (DSS) information based on the work of Liu, et al. 2018 [23]. Survival utilizes R packages survival (v2.41-3) and survminer (v0.4.2). The optimal-cut points based on the lowest log-rank *p*-value, with the minimum proportional comparison set to 15% vs. 85% of samples, were determined the “surv_cutpoint” function in the survminer package. Hazard ratio (HR) are derived from the Cox Proportional Hazard regression model and are based on the high-versus-low comparison.

### 2.5. SETD2 Gene Expression Analysis in TCGA PRAD Database

To explore the relative *SETD2* overexpression and associated gene set enrichment data, we sourced the Cancer Genome Atlas Prostate Adenocarcinoma (TCGA PRAD) database. Differential gene expression analysis was generated using the Biolake online web-tool available at biolake.ucalgary.ca (accessed on 26 December 2023) [24].

### 2.6. Statistical Analysis

Statistical analysis for our study cohort was performed using SPSS (version 25.0, IBM Corp., Armonk, NY, USA). Frequency and proportions were reported for categorical data. Mean and standard deviations (SDs) were reported for normally distributed continuous data; median and range were reported for non-normally distributed continuous data. Chi-square tests were used to compare two categorical variables, and Fisher’s exact test was used where the cell frequencies were <5. Overall survival (OS), defined as the time from the date of diagnosis of PCa as detected on a TURP specimen to the date of death, and prostate cancer-specific survival (CSS), defined as death due to PCa, were analyzed using the Kaplan–Meier method. The median time and the 95% confidence interval (CI) were reported. Log rank tests were used to compare two or more survival curves. Cox proportional hazards models were used to determine the factors associated with OS and CSS; the hazard ratio (HR) and the corresponding 95% CI were reported. The adjusted Cox model was fitted as well. A *p* value of <0.05 was used for statistical significance, and two-sided tests were used.

## 3. Results

### 3.1. SETD2 Expression in the Prostate Cancer Cohort

Our results showed that all cases of prostate cancer showed differentially expression of SETD2 (Figure 1A). The mean intensity of advanced and castrate-resistant disease was significantly higher than incidental cases (2.53 ± 0.58 and 2.21 ± 0.63 vs. 1.9 ± 0.68; *p* < 0.001). Interestingly, we observed a significant difference when comparing the mean intensity for deceased vs. alive patients (2.31 ± 0.66 vs. 2 ± 0.68; *p* = 0.003), respectively (Figure 1B–C). SETD2 high expression (score 3) was found to be significantly associated with both OS and CSS, *p*-values of 0.001 and 0.0001, respectively (Figure 1D,E).

### 3.2. SETD2 Expression in Relation to Gleason Grade Grouping

There was a significant association between high SETD2 expression and Gleason grade groups (*p* < 0.0001). In this cohort, 42/78 (53%) of GG5 showed SETD2 intensity score 3, or high expression, vs. 51/63 (81%) of GG1 showing SETD2 intensity score 1, 2, or low risk, as shown in Table 2.

### 3.3. SETD2 Expression in Relation to PTEN, ERG, and p53

The association between low-risk/high-risk SETD2 IHC expression groups compared to ERG, PTEN, TP53, and AR (low and high risk groups) is described in Table 2. Our data revealed that PTEN loss of expression was observed in 31 (44.9%) in the SETD2 high-risk group versus 24 (20.9%) in the SETD2 low-risk group, *p* = 0.0001. The high-risk SETD2 group had more PTEN loss tumors (*n* = 31; 44.9%, *p* = 0.0001) than the low-risk SETD2 group (*n* = 24; 20.9%, *p* = 0.0001). In TP53 abnormal cases relative to low and high-risk SETD2 groups, TP53 abnormality was observed in 26.4% versus 10.2% of high and low SETD2 (*p* < 0.001), respectively.

When we compared SETD2 high-/low-risk groups to ERG-positive tumors, they were more frequently observed among high-risk SETD2 vs. low-risk SETD2 tumors but not significant (33.3% versus 23.3%; *p* = 0.136). Similarly, AR was noy significant when compared to the high- or low-risk groups.

### 3.4. High SETD2 Expression Is Associated with Poor Overall Survival (OS) and Cause-Specific Survival (CSS) Related to Prostate Cancer Lethality

We extended our investigation to evaluate SETD2 expression in relation to patients’ survival, such as OS and CSS. Overall, our data indicated that high-risk SETD2 expression tumors were associated with lower OS (HR 1.80; 95% CI: 1.28–2.53, *p* = 0.001) and lower CSS (HR 3.14; 95% CI: 1.94–5.08, *p* < 0.0001) (Table 3).

The distribution of SETD2 high-risk and PTEN and ERG biomarkers in our cohort was evaluated using univariate and multivariate analysis, as depicted in Table 3. In summary, PTEN loss combined with either high- or low-risk SETD2 showed worse OS and CSS. However, when adjusted for the Gleason grade group, the only significant combination was that of PTEN loss with high-risk SETD2, which supported the prognostic value of combining those two biomarkers (Table 3).

The same trend was also observed when combining SETD2 with ERG status. Specifically, high-risk SETD2 association with ERG positive cases revealed poorer OS and CSS survival curves, with 95% CI and HR = 3.30 (1.96–5.58, *p* < 0.0001) and HR = 7.64 (3.59–16.25, *p* < 0.0001) for OS and CSS for PCa, respectively. When adjusted for Gleason grade groups, high SETD2/ERG positivity was again the only combination showing statistical significance vs. any other combination for OS and CCS, respectively (HR 1.99, 95% CI 0.87–4.59; *p* = 0.015) and (HR2.14, 95% CI 0.98–4.68, *p* = 0.058), respectively (Figure 2). Our data did not demonstrate a significant impact on survival when the combination of TP53 abnormalities and SETD2 expression was investigated

### 3.5. High Expression of SETD2 Is a Risk Factor and an Indicator of Poor Prognosis in Prostate Cancer (PCa)

TCPA protein expression and associated clinical data analysis revealed SETD2 high expression is associated with poor prognosis in PCa samples OS (HR: 13.6) and DFS (HR: 7.96) (Figure 3A,B). Furthermore, when the Cox regression of the hazard ratio of all differentially expressed proteins is performed, SETD2 appears at the top of the list of poor prognosis as shown in Figure 3C. We further looked at the proportion of dysregulated proteins and plotted the proteins that indicated poor and good prognosis. Our data showcased SETD2 as one of the poor prognosis markers using TCPA data (Figure 3D).

### 3.6. Gene Set Enrichment Analysis of the Revealed Potential Role of SETD2 in PCa Oncogenesis

TCGA PRAD data analysis revealed poor overall and progress-free survival associated with upregulated *SETD2* gene expression (Figure 4A). Moreover, our data shows a dramatic increase in *SETD2* gene expression with an increased Gleason score (Figure 4B). Differential gene expression analysis was performed, and a volcano plot of log 2 (fold changes) was blotted (Figure 4C). Furthermore, we obtained the positive and negative correlated gene lists and the top 50 genes presented in Figure 4D. The gene set enrichment analysis (GSEA) revealed potential enrichments associated with SETD2 gene expression, including the AMPK signaling pathway, the cAMP signaling pathway, and the PI3K-Akt signaling pathway, which are potentially associated with tumor progression, chemoresistance, and a poor prognosis (Figure 5).

## 4. Discussion

We explored whether the involvement of SETD2 in PCa may provide valuable insights as a prognostic marker. In the current study, elevated SETD2 expression was significantly associated with a poorer prognosis, indicating its significance as a potential biomarker for disease aggressiveness and lethality. Mutations in the SETD2 gene were reported in many tumors, leading to more aggressive biological behavior of the tumor, including features related to its plasticity [25,26].

Our results demonstrate that SETD2 high-risk (score 3) was associated with disease progression and dramatically impacted OS and CSS. This finding was also significant when combined SETD2 expression with PTEN loss or ERG gain provided more prognostic value for predicting OS and CSS compared to SETD2 alone. It is well known that PTEN loss leads to hyperactivation of the PI3K/AKT/mTOR pathway, promoting cell survival, proliferation, and metastasis [27]. On the other hand, ERG rearrangements are often associated with a poor prognosis and aggressive disease [28]. The combined effect of SETD2 overexpression with ERG gain or PTEN loss may lead to more aggressive tumor behavior, including increased metastatic potential and resistance to treatment, resulting in poorer prognostic outcomes for patients [29]. These findings support the potential role of SETD2 as an oncogene, as evidenced by the higher prognostic values when combining risk with the expression of SETD2 with PTEN or ERG.

Previous studies investigating the *SETD2* mutation and its association with tumor burden have yielded conflicting findings and showcased the complexity of their roles in disease progression. In fact, similar observations using TCGA data show a differential expression pattern of the *SETD2* gene in various tumors. Despite these discrepancies, these studies provide valuable insights into the molecular mechanisms underlying tumor development and progression. The reasons behind these contradictory results could be explained by variations in patient cohorts, tumor sites, and heterogeneity.

It is worth nothing that investigations into the downstream signaling pathways of the SETD2 GSEA (Figure 5) could elucidate critical molecular drivers of aggressive prostate cancer phenotypes. The GSEA analysis of TCGA PRAD data highlighted the *SETD2* gene as potentially playing a pivotal role in several cellular pathways, notably impacting the AMPK, cAMP, and PI3K-Akt signaling pathways, contributing to tumor progression, and fostering prostate cancer cell proliferation, invasion, and metastasis [30]. *SETD2* involvement in these pathways influences crucial cellular processes like metabolism, growth, and survival, potentially leading to chemoresistance. Integrating multi-omics approaches and advanced computational analyses may help solve the conflicting findings. But further in vitro and in vivo models are needed to validate the molecular mechanism.

Even though different expression trends were observed in other tumor sites, such as kidney, gastric, or pancreatic tumors, to overcome these, we explored the PanCancer database for gene expression. We found *SETD2* overexpression was significantly higher in acute myeloid leukemia (AML), esophagus cancer, and pancreatic cancer compared to normal. Also, upregulation was observed in liver, stomach, and renal cancer but was statistically not significant. Interestingly, in TCPA data, SETD2 was one of the top proteins with the highest HR in the prostate cohort and was considered a prognostic marker.

In our data, we identify the high-risk group of SETD2 expression in PCa. While we rely on small cohorts, this approach presents several limitations. Small sample sizes and heterogeneity of the disease, leading to biased or inconclusive results. To overcome this limitation, we recommend wide-scale analyses involving larger cohorts and integrating data from multi-center studies to increase the statistical power and robustness of the findings. Alternatively, employing state-of-the-art techniques such as single-cell sequencing or spatial transcriptomics could offer deeper insights into the spatial and temporal dynamics of protein expression within the tumor microenvironment.

## 5. Conclusions

Our data suggest a prognostic role for SETD2 in a prostate cancer cohort. High SETD2 expression is associated with disease lethality, especially in advanced and castrate-resistant disease cases, and high-risk SETD2 expression revealed poor prognosis as well as worse OS and CSS. Additionally, the combination of high SETD2 expression with PTEN loss or ERG-positive expression signifies poor outcomes. The protein expression database (TCPA) also indicates that SETD2 expression is associated with a poor prognosis.

## Figures and Tables

**Figure 1 cancers-16-01436-f001:**
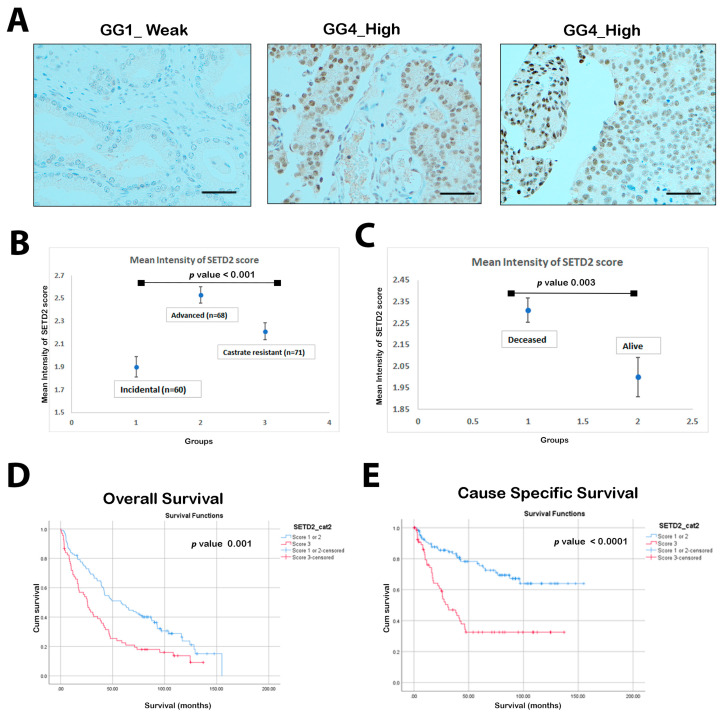
SETD2 expression is associated with poor overall survival and cause-specific survival. (**A**) Immunohistochemistry (IHC) assessment for SETD2 in prostate tissues with weak to high expression patterns. Top panel showing Acinar Adenocarcinoma Gleason score 3 + 3 (GG1), second IHC image Gleason score 4 + 4 (GG4), moderate to high nuclear stain. Right top IHC image: strong nuclear Gleason score 4 + 4 (GG4). (Scale bar = 100 μm). (**B**) Boxplots show the expression intensity of SETD2 (mean ± SD) in incidental, advanced, and castrate-resistant PCa tissues. SETD2 protein expression levels were scored through IHC. Each sample was scored semi-quantitatively using a three-tiered system (weak—1; moderate—2; and strong—3). The error bars indicate the standard error of the mean. *p*-value < 0.001. (**C**) Box plot representing the BAP1 intensity (mean ± SD) compared patients to survival status (deceased and live) (*p* value = 0.003). (**D**) Kaplan–Meier (KM) curve representing overall survival (OS) on IHC expression of SETD2 in relation to intensity groups: low-risk (scores 1 and 2) and high-risk (score 3). (**E**) KM curves representing the cause-specific survival (CSS) on IHC expression of SETD2 in relation to intensity groups: low-risk (scores 1 and 2) and high-risk (score 3).

**Figure 2 cancers-16-01436-f002:**
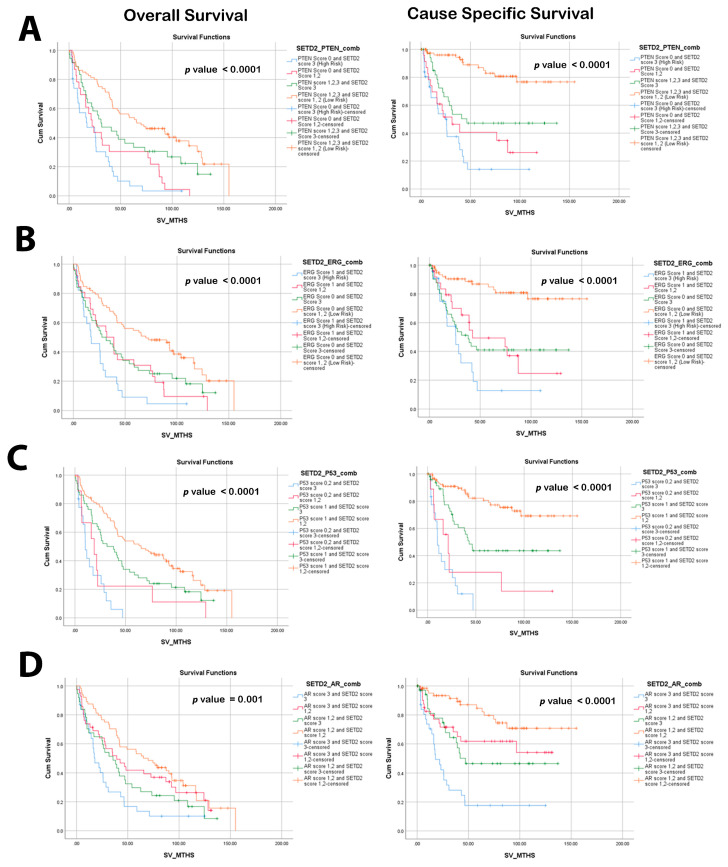
Kaplan–Meier survival curve illustration of overall survival (OS) and cause-specific survival (CSS) SETD2 among low- and high-risk groups, in combination with (**A**) PTEN scores and (**B**) ERG. (**C**) TP53, and (**D**) AR mutation status.

**Figure 3 cancers-16-01436-f003:**
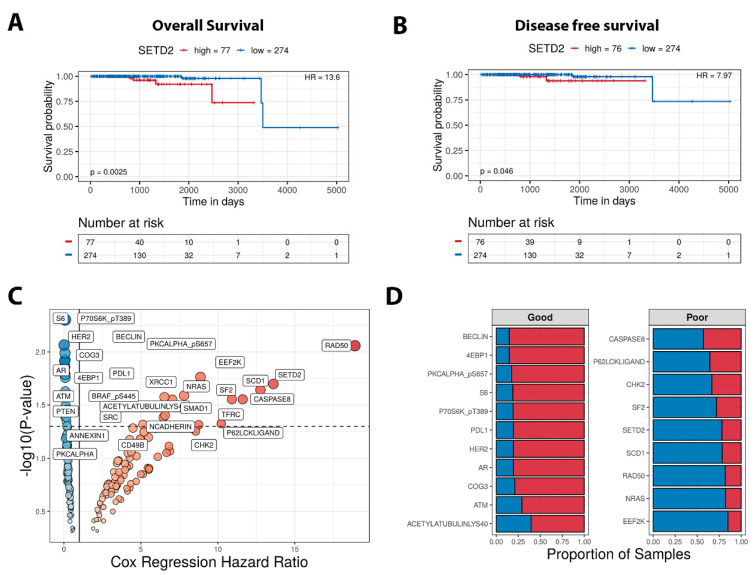
SETD2 protein expression is an indicator of poor prognosis. (**A**) Kaplan–Meier (KM) overall survival analysis of SETD2 expression and its association with PCa survival probability. (**B**) KM curves representing SETD2 disease-free survival (DFS). (**C**) Volcano plots showing the log10 (*p* values) of each protein across PCa and Cox regression of the hazard ratio. (**D**) Good and poor prognostic markers in TCPA data of PCa samples.

**Figure 4 cancers-16-01436-f004:**
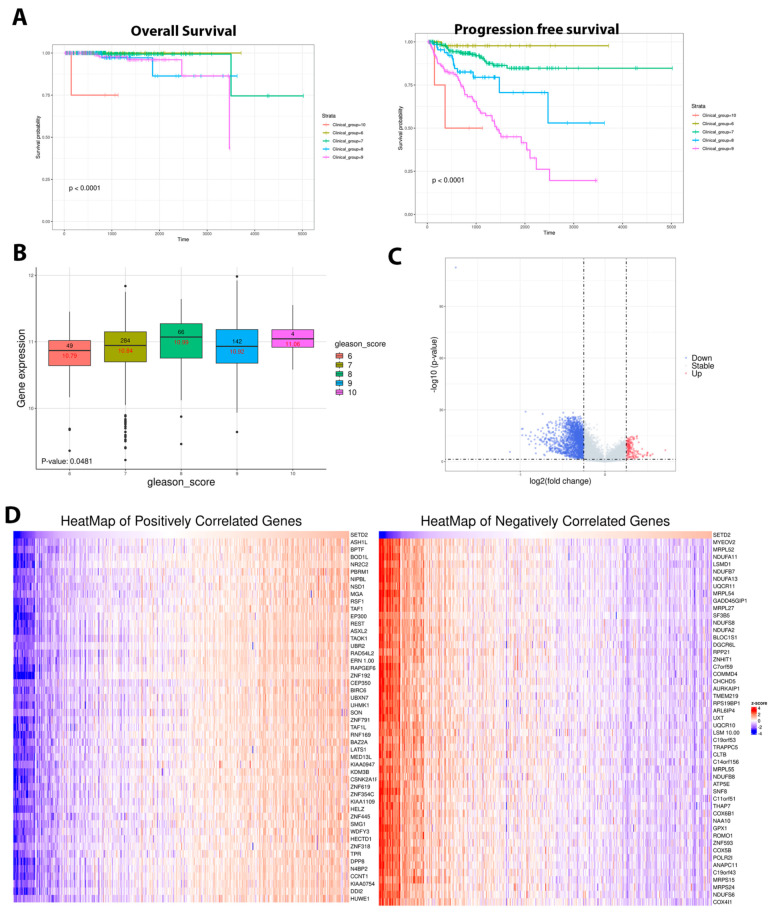
SETD2 gene expression using TCGA PRAD data analysis. (**A**) Kaplan–Meier curves of overall and progression-free survival analysis of SETD2 gene expression in PCa patients associated with clinical group (Gleason score). (**B**) Box plot showing the SETD2 expression and Gleason score; (**C**) Volcano plot representing differential gene expression in TCGA PRAD associated with SETD2; (**D**) Heatmaps showing the top 50 positive and negative correlated genes.

**Figure 5 cancers-16-01436-f005:**
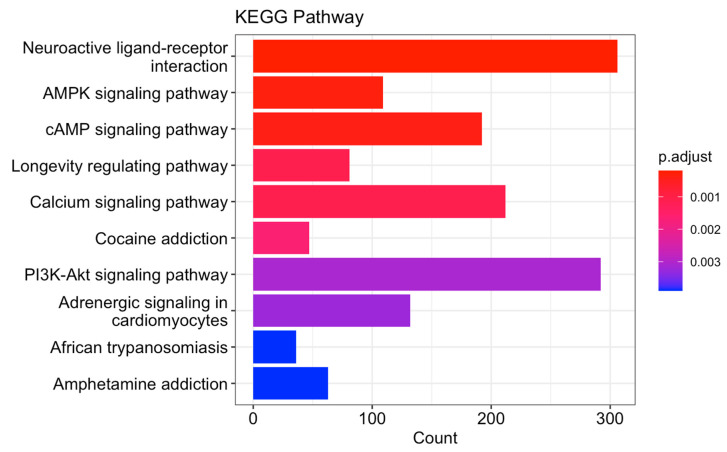
Gene set enrichment analysis showing the top KEGG GO terms associated with SETD2 gene overexpression.

**Table 1 cancers-16-01436-t001:** Patient demographics (*n* = 202).

ISUP Grade Group Cohort (Gleason Score)	Number and Percentage %
Grade group 1	63 (61.1%)
Grade group 2 (3 + 4)	14 (13.8%)
Grade group 3 (4 + 3)	20 (19.5%)
Grade group 4 (8)	11 (14.6%)
Grade group 5 (9,10)	78 (91%)
Missing	16 (7.9%)
Deceased	
Yes	139 (67.8%)
No	57 (28.2%)
Missing	8 (3.9%)
Cancer subgroup	
Incidental	61 (30.2%)
Advanced	69 (34.2%)
Castrate-resistant	72 (35.6%)
SEDT2 score (by cancer subgroup)	
Score 1	
Incidental	17 (60.7%)
Advanced	3 (10.7%)
Castrate-resistant	8 (28.6%)
Score 2	
Incidental	32 (32.7%)
Advanced	26 (26.5%)
Castrate-resistant	40 (40.8%)
Score 3	
Incidental	11 (15.1%)
Advanced	39 (53.4%)
Castrate-resistant	23 (31.5%)
TP53 and SEDT2 combined	
TP53 scores 0, 2, 3, and SEDT2 1/2	0 (0%)
TP53 scores 0, 2, 3, and SEDT2 3	31 (18.1%)
TP53 score 1 and SEDT2 1/2	25 (100%)
TP53 score 1 and SEDT2 3	134 (78.6%)
PTEN and SEDT2 combined	
PTEN loss and SEDT2 score 1/2	3 (12%)
PTEN loss and SEDT2 score 3	52 (30.4%)
PTEN intact and SEDT2 1/2	22 (88%)
PTEN intact and SEDT2 3	107 (62.5%)
ERG and SEDT2 combined	
ERG-positive and SEDT2 score 1/2	27 (23.3%)
ERG-positive and SEDT2 score 3	23 (33.3%)
ERG-negative and SEDT2 score 1/2	89 (76.7%)
ERG-negative and SEDT2 score 3	46 (66.7%)

SETD2 is scored using a four-tiered system (0, 1, 2, and 3 as no expression, weak, moderate, and high, respectively). PTEN (score 0) = negative staining = PTEN loss (high risk). PTEN scores 1, 2, and 3 indicate weak, moderate, or high staining (low risk), respectively. ERG-risk groups: positive—gain (high risk); negative—loss (low risk). TP53 score: 0 = normal; TP53 scores 0, 2, and 3 = mutant.

**Table 2 cancers-16-01436-t002:** Comparison between SETD2 low/high risk and Gleason score, and common genomic alterations status.

Variable	SETD2 Score 3 (High Risk)	SETD2 Score 1 or 2 (Low Risk)	*p*-Value
Gleason acore			
<=6	12 (17.1)	51 (44.0)	<0.0001
3 + 4	3 (4.3)	11 (9.5)	
4 + 3	4 (5.7)	16 (13.8)	
8	9 (12.9)	2 (1.7)	
9–10	42 (60.0)	36 (31.0)	
PTEN intensity			
Score 0	31 (44.9)	24 (20.9)	0.001
Score > 0	38 (55.1)	91 (79.1)	
ERG dual intensity			
Negative	46 (66.7)	89 (76.7)	0.136
Positive	23 (33.3)	27 (23.3)	
AR			0.237
Score 1,2	39 (54.2)	76 (62.8)	
Score 3	33 (45.8)	45 (37.2)	
P53			0.003
Score 1	53 (73.6)	106 (89.8)	
Score 0, 2	19 (26.4)	12 (10.2)	

**Table 3 cancers-16-01436-t003:** Univariate and multivariate analysis of survival data associated with SETD2 expression groups.

	Overall Survival HR (95% CI)	*p*-Value	Cause-Specific Survival HR (95% CI)	*p*-Value
PTEN gain score 1, 2, and 3)				
Loss—score 0	2.71 (1.89–3.89)	<0.0001	4.50 (2.76–7.34)	<0.0001
ERG (Negative)				
Positive	1.93 (1.34–2.77)	<0.0001	2.53 (1.55–4.11)	<0.0001
GS (<=6)				
GS 3 + 4	2.03 (0.98–4.20)	0.056	24.18 (2.70–216.62)	<0.0001
GS 4 + 3	1.41 (0.72–2.77)	0.322	9.68 (1.01–93.15)	0.049
GS 8	5.09 (2.47–10.49)	<0.0001	71.43 (8.68–588.10)	<0.0001
GS 9, 10	6.02 (3.80–9.51)	<0.0001	107.23 (14.73–780.83)	<0.0001
SETD2 low risk—score 1 or 2				
SETD2 high risk—score 3	1.80 (1.28–2.53)	0.001	3.14 (1.94–5.08)	<0.0001
Combination PTEN and SETD2 (PTEN score 1, 2, and 3 and SETD2 score 1 and 2)				
PTEN score 0 and SETD2 score 3	3.78 (2.36–6.08)	<0.0001	10.10 (4.93–20.67)	<0.0001
PTEN score 0 and SETD2 score 1 and 2	2.68 (1.62–4.41)	<0.0001	6.59 (3.07–14.12)	<0.0001
PTEN score 1, 2, and 3 and SETD2 score 3	1.60 (1.01–2.54)	0.047	3.98 (1.90–8.35)	<0.0001
Combination PTEN and SETD2 (PTEN score 1, 2, and 3 and SETD2 score 1 and 2) *				
PTEN score 0 and SETD2 score 3	2.12 (1.22–3.69)	0.008	3.74 (1.67–8.34)	0.001
PTEN score 0 and SETD2 score 1 and 2	1.28 (0.73–2.26)	0.395	1.85 (0.81–4.22)	0.144
PTEN score 1, 2, and 3 and SETD2 score 3	0.96 (0.58–1.59)	0.866	1.55 (0.70–3.43)	0.278
Combination ERG and SETD2 (ERG negative and SETD2 score 1 and 2)				
ERG positive and SETD2 score 3	3.30 (1.96–5.58)	<0.0001	7.64 (3.59–16.25)	<0.0001
ERG positive and SETD2 score 1 and 2	1.99 (1.22–3.25)	0.006	4.20 (1.97–8.96)	<0.0001
ERG negative and SETD2 score 3	1.80 (1.17–2.76)	0.008	4.41 (2.24–8.67)	<0.0001
Combination ERG and SETD2 (ERG negative and SETD2 score 1 and 2) *				
ERG positive and SETD2 score 3	1.99 (0.87–4.59)	0.015	2.14 (0.98–4.68)	0.058
ERG positive and SETD2 score 1 and 2	1.45 (0.73–2.89)	0.292	1.12 (0.50–2.51)	0.782
ERG negative and SETD2 score 3	1.02 (0.63–1.63)	0.943	1.54 (0.77–3.11)	0.225

* Adjusted for gleason score.

## Data Availability

The data can be shared upon request.

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
