# Peer review of "Exploring The Prognostic Significance of SET-Domain Containing 2 (SETD2) Expression in Advanced and Castrate-Resistant Prostate Cancer"

_cancers, 2024, doi:10.3390/cancers16071436_

Round 1

Reviewer 1 Report

Comments and Suggestions for Authors

The manuscript investigates the potential role of SETD2, a gene involved in epigenetic regulation, as a biomarker in prostate cancer (PCa). The study involves a cohort of 202 PCa patients treated non-surgically with androgen deprivation therapy. They discovered that SETD2 expression was associated with advanced disease and higher mortality. Additionally, there was a significant correlation between SETD2 expression and Gleason grade groups. Association with other biomarkers (PTEN, ERG, p53) revealed potential combinatorial prognostic value. Overall, the manuscript contributes valuable insights into the potential clinical relevance of SETD2 in prostate cancer. To provide a more comprehensive understanding, the following analysis could be incorporated:

1. Figures and data presentation need to be improved for better visual representation. The font format also need to be consistent.

2. If applicable, any validation procedures could be employed for the immunohistochemistry results to ensure the reliability of SETD2 expression assessments.

3. If applicable, functional studies on SETD2 could be conducted to understand how SETD2 alterations influence cellular processes, metastasis, and response to therapies.

4. Any genomics, transcriptomics, and proteomics data could be integrated to gain a more holistic understanding of the molecular landscape associated with SETD2 and prostate cancer.

5. Any correlation between SETD2 expression and immune cell infiltration in the tumor microenvironment.

Minors:

1. Missing p values in Figure 1B,C,D,E, and Figure 2.

Author Response

Reviewer 1

The manuscript investigates the potential role of SETD2, a gene involved in epigenetic regulation, as a biomarker in prostate cancer (PCa). The study involves a cohort of 202 PCa patients treated non-surgically with androgen deprivation therapy. They discovered that SETD2 expression was associated with advanced disease and higher mortality. Additionally, there was a significant correlation between SETD2 expression and Gleason grade groups. Association with other biomarkers (PTEN, ERG, p53) revealed potential combinatorial prognostic value. Overall, the manuscript contributes valuable insights into the potential clinical relevance of SETD2 in prostate cancer. To provide a more comprehensive understanding, the following analysis could be incorporated:

# Response: We appreciate the reviewer's feedback and constructive suggestions to enhance the readability and coherence of the manuscript. We extensively edited and refined the revised manuscript version, including thorough proofreading by experts. We apologize for any shortcomings in the clarity of previous version writings. We are committed to addressing all concerns raised and improving the overall quality of the paper to ensure it meets the highest standards of the journal and scientific communication.

Figures and data presentation need to be improved for better visual representation. The font format also need to be consistent.

# Response: Thank you. The figures were corrected and replaced with high resolution as per your suggestion.

2. If applicable, any validation procedures could be employed for the immunohistochemistry results to ensure the reliability of SETD2 expression assessments.

# Response: IHC validation was conformed using multi-tissue control tissue microarray slide containing samples of multi organ cancers/benign samples was screened using the primary antibody or normal mouse 1:200 prediluted serum (Ventana Medical Systems, Inc.) as positive and negative controls, respectively.

3. If applicable, functional studies on SETD2 could be conducted to understand how SETD2 alterations influence cellular processes, metastasis, and response to therapies.

# Response: Thank you for your suggestion. However, given that our manuscript is mainly investigating prognostic value for SETD2 and its association with other biomarkers, the functional studies suggested will take tremendous time and is out of scope of this current study. They could be suitable for subsequent investigational study about the role of SETD2 in progression and metastasis. We acknowledge the significance of such suggestion. However, our study primarily centered around exploring the outcomes of

overall survival (OS) and cancer-specific survival (CSS), as well as investigating the synergistic impact of lethal molecular genetic alterations on PCa lethality.

4. Any genomics, transcriptomics, and proteomics data could be integrated to gain a more holistic understanding of the molecular landscape associated with SETD2 and prostate cancer.

# Response: We included analysis of both TCGA and TCPA Proteomics data from publicly available database for more holistic understanding of the molecular landscape associated with SETD2 and prostate cancer.

5. Any correlation between SETD2 expression and immune cell infiltration in the tumor microenvironment.

# Response: Immune cell infiltration in the tumor microenvironment is important and wide area for exploration. Since we focused on outcomes and correlation with genomic alteration. So, we didn’t look at immune cell infiltration in the tumor microenvironment area as its will be different topic.

Minors:

1. Missing p values in Figure 1B, C, D, E, and Figure 2.

# Response: p values were added as per reviewer recommendation.

Reviewer 2 Report

Comments and Suggestions for Authors

The research conducted by Yaser Gamallat and colleagues has significant shortcomings. The manuscript is poorly written, riddled with grammatical errors, lacks coherent logic, and the discussion is utterly nonsensical.

Major comments:

1.     Some sentences are very difficult to understand. Such as “These functions can be achieved by H3K36me3 ability of recruiting protein complexes that work on DNA repair such as: DNA methylation, histone methylation and acetylation, DNA binding, and DNA mismatch repair; but also by H3K36me3 role on fostering transcriptional protection; to restore chromatin function after transcription, through transcriptome elongation, interaction with RNA polymerase II, and mismatch repair.”, “Additionally, the discovery of SEDT2-AMPK-EZH2 path-ways on PCa can provide new therapeutics options, which can be beneficial to patients that developed castration resistance or androgen receptor indifferent forms of disease . Metformin induces SEDT2 activity [29].” et al. Please re-write the manuscript.

2.     Really surprised me that there’s not even one sentence relative to Figures 2 and 5.

3.     All the labels in Figure 4 are confusing. The image resolution is very bad.

4.     In citation 28, Yuan, H., et al. reported SETD2 restricts prostate cancer metastasis, which conclusion is opposite to the conclusion from this research. The author should discuss the reason for this inconsistency.

Comments on the Quality of English Language

Very bad.

Author Response

The research conducted by Yaser Gamallat and colleagues has significant shortcomings. The manuscript is poorly written, riddled with grammatical errors, lacks coherent logic, and the discussion is utterly nonsensical.

# Response: We appreciate the reviewer's feedback on our manuscript. We extensively and thoroughly edit and refine the revised manuscript, including thorough proofreading by experts. We apologize for any shortcomings in the clarity of previous version writings. We revised the manuscript and considered enhancement of both readability and coherence of the manuscript. We also committed to addressing all concerns raised and improving the overall quality of the paper to ensure it meets the standards of scientific communication.

Major comments:

1. Some sentences are very difficult to understand. Such as “These functions can be achieved by H3K36me3 ability of recruiting protein complexes that work on DNA repair such as: DNA methylation, histone methylation and acetylation, DNA binding, and DNA mismatch repair; but also by H3K36me3 role

on fostering transcriptional protection; to restore chromatin function after transcription, through transcriptome elongation, interaction with RNA polymerase II, and mismatch repair.”, “Additionally, the discovery of SEDT2-AMPK-EZH2 path-ways on PCa can provide new therapeutics options, which can be beneficial to patients that developed castration resistance or androgen receptor indifferent forms of disease . Metformin induces SEDT2 activity [29].” et al. Please re-write the manuscript.

# Response: Thank you for your valuable feedback. We apologize for any shortcomings. The revised manuscript was improved for better readability and clarity.

2. Really surprised me that there’s not even one sentence relative to Figures 2 and 5.

# Response: Thank you. Proper citation added.

3. All the labels in Figure 4 are confusing. The image resolution is very bad.

# Response: We apologize for any inconvenience caused by the poor quality of the images. We uploaded high-resolution images in the revised manuscript.

4. In citation 28, Yuan, H., et al. reported SETD2 restricts prostate cancer metastasis, which conclusion is opposite to the conclusion from this research. The author should discuss the reason for this inconsistency.

 # Response: Thank you. Our cohort was not comprised by patients with metastatic prostatic cancer at diagnosis. Samples collected were all from prostate primary tumors, and not from distant metastasis. Patients had either incidental, castrate-resistant, or advanced disease. SETD2 expression has been showing variation among tumors, in our cohort, overexpression or high-risk pattern was associated to worse survival outcomes. The citation was precise and was focused on showing how abnormal SEDT2 functions (in Yuan’s study showing under expression) in metastatic prostate carcinoma, opening a window of opportunity to explore new therapeutic approaches.

Reviewer 3 Report

Comments and Suggestions for Authors

Yaser et al. investigate the prognostic role of SETD2, a histone methyltransferase, in a prostate cancer cohort. The results show that high SETD2 expression is associated with advanced and castrate-resistant disease, as well as poor prognosis and worse overall survival. Additionally, the combination of high SETD2 expression with PTEN loss or ERG positivity is also associated with poor outcomes. The protein expression database (TCPA) also indicates that SETD2 expression is associated with a poor prognosis.

In general, the results shown in this manuscript are good, and the topic is intriguing, but it is poorly prepared. I suggest the authors revise the manuscript by majority.

Here are some major comments:

  1. The resolution of the photos in the manuscript is inadequate. Additionally, the table's format does not adhere to the guidelines. Specifically, Table 2 appears to be a screenshot from an Excel spreadsheet. The manuscript's figures and tables are not meant for PPT presentation, the authors should make that clear. Chart resolution, format, and size ought to be uniform. When reviewing the document, there are numerous challenges due to the low image resolution, and certain important legends need to be inferred. Certain passages of the text have strange writing styles as well. Additionally, the author should be aware of international writing customs while using English, such as "rearrangments," where the author omits the letter E following the letter G in the fourth paragraph of result 3.4. Even though some nations and areas have this writing style, it still needs to follow the conventions of the majority of professional readers. In addition, some words have the wrong spelling. For instance, "ethial" should read "ethical" in the third sentence of the fourth paragraph of Materials and Methods Section 2.1. Additionally, there are instances in the manuscript when punctuation is absent, including at the conclusion of the introduction's second paragraph.

  1. There is an issue with the introduction's logic. There isn't enough discussion of why SETD2 was chosen as the study subject. Biomarkers associated with genetic aberrations ought should be the main focus of attention, based on the second paragraph of the introduction; yet, the discussion that follows does not address biomarkers associated with genetic aberrations. The SET2 gene is explicitly introduced in the third paragraph of the introduction; nevertheless, no transitional relationship is established between it and genetically linked biomarkers. It appears quite sudden. Furthermore, the outcomes of tumor biology study connected to H3K36me3 are covered in full in the third paragraph. This passage makes no connection between the expression of SETD2 or genetic alterations and the function of H3K36me3, despite the fact that the SETD2 gene is the sole gene for H3K36me3 methylation events that has been written down. To put it succinctly, in order to fully justify the SETD2 gene as a research object, the introduction's logic must be reorganized and its structure must be rebuilt. Additionally, keep in mind that the entire work consists of just 31 references. In this manuscript, references 3, 4, 5, 18, 19, and 29 are the six citations for the author's own published essay. The percentage of self-citations was 19.35%. In my opinion, the author's self-citation contains pertinent findings that could be supported by more materials. For the purpose of lowering the self-citation rate, authors ought to cite published material more frequently.

  1. The authors did not go into detail regarding SETD2 protein expression scoring in Materials and Methods Section 2.3. The scoring standards are merely stated. More specifically, how are samples with high expression levels found? Is a field of view considered to have a high-expression sample if it contains a significant number of positively stained cells, or is there another way? Put differently, how do you handle inconsistent expressions in the field of view while scoring? I think that in real testing, this is a common occurrence. For instance, all positive cells exhibit inconsistently high staining signals in Figure 1A, which displays the IHC staining data for GG4_High. Furthermore, the author must include trustworthy sampling information, such as the number of sections that were obtained for each sample's IHC staining, the number of fields of view that were examined, etc. It is crucial to consider the number of slices and viewing windows. Because the relative density score of the SETD2 expression is essentially around 2, as result 3.1 shows. The sample size affects the statistical significance between the comparison groups in Figures 1B and 1C.

  1. Consequently, Part 3.6's conclusion is incredibly weak. Figure 4A indicates that there are variations in survival between various clinical groups, based on the legend. However, it is impossible to determine whether the variations in survival across various clinical groups are statistically different because significance testing is not used. Furthermore, there is no connection between the expression of SETD2 and Figure 4A. However, Figure 4B does not have significance testing, so it is not possible to conclude that SETD2 expression rises as the Gleason score increases. Additionally, this section only summarizes the findings of several analyses with reference to enrichment analysis. The author jumps to conclusions without bothering to connect the observable findings to particular occurrences. For instance, according to the authors, SETD2 expression is linked to treatment resistance and a dismal prognosis. I'm not sure how such a determination is made.

In addition, a few small recommendations and queries are stated below.

·      As may be seen in Figures 1D and 1E, the author combined Scores 1 and 2 into a single group. Why is it done in this manner? It is not stated whether this treatment is necessary. Why not combine points two and three?

·      Markers for the significance test ought to be added to Figures 1B and 1C. Analogously, significance testing is also necessary for Figures 1D and 1E.

·      Does the description in the results section 3.2 correspond to the results in Table 2?

·      The explanations of the two results are virtually duplicated in the first paragraph of Results 3.3. In essence, Table 3 is Figure 2's data table format.

·      Why is the analysis based on the TCPA data set not using the four-tiered system? Rather, the expression is merely separated into categories based on high and low expression.

Comments on the Quality of English Language

There are errors in the use of punctuation marks and spelling of words.

Author Response

Yaser et al. investigate the prognostic role of SETD2, a histone methyltransferase, in a prostate cancer cohort. The results show that high SETD2 expression is associated with advanced and castrate-resistant disease, as well as poor prognosis and worse overall survival. Additionally, the combination of high SETD2 expression with PTEN loss or ERG positivity is also associated with poor outcomes. The protein expression database (TCPA) also indicates that SETD2 expression is associated with a poor prognosis.

In general, the results shown in this manuscript are good, and the topic is intriguing, but it is poorly prepared. I suggest the authors revise the manuscript by majority.

Here are some major comments:

1. The resolution of the photos in the manuscript is inadequate. Additionally, the table's format does not adhere to the guidelines. Specifically, Table 2 appears to be a screenshot from an Excel spreadsheet. The manuscript's figures and tables are not meant for PPT presentation, the authors should make that clear. Chart resolution, format, and size ought to be uniform. When reviewing the document, there are numerous challenges due to the low image resolution, and certain important legends need to be inferred. Certain passages of the text have strange writing styles as well. Additionally, the author should be aware of international writing customs while using English, such as "rearrangments," where the author omits the letter E following the letter G in the fourth paragraph of result 3.4. Even though some nations and areas have this writing style, it still needs to follow the conventions of the majority of professional readers. In addition, some words have the wrong spelling. For instance, "ethial" should read "ethical" in the third sentence of the fourth paragraph of Materials and Methods Section 2.1. Additionally, there are instances in the manuscript when punctuation is absent, including at the conclusion of the introduction's second paragraph.

# Response: Thank you for bringing this to our attention, and we are committed to provide clear and visually accessible figures and enhance the overall quality of the manuscript. We apologize for any inconvenience caused by the poor quality of the images. We have intensively edited the manuscript and uploaded high-resolution images in the revised manuscript.

2. There is an issue with the introduction's logic. There isn't enough discussion of why SETD2 was chosen as the study subject. Biomarkers associated with genetic aberrations ought should be the main focus of attention, based on the second paragraph of the introduction; yet, the discussion that follows does not address biomarkers associated with genetic aberrations. The SET2 gene is explicitly introduced in the third paragraph of the introduction; nevertheless, no transitional relationship is established between it and genetically linked biomarkers. It appears quite sudden. Furthermore, the outcomes of tumor biology study connected to H3K36me3 are covered in full in the third paragraph. This passage makes no connection between the expression of SETD2 or genetic alterations and the function of H3K36me3, despite the fact that the SETD2 gene is the sole gene for H3K36me3 methylation events that has been written down. To put it succinctly, in order to fully justify the SETD2 gene as a research object, the introduction's logic must be reorganized and its structure must be rebuilt. Additionally, keep in mind that the entire work consists of just 31 references. In this manuscript, references 3, 4, 5, 18, 19, and 29 are the six citations for the author's own published essay. The percentage of self-citations was 19.35%. In my opinion, the author's self-citation contains pertinent findings that could be supported by more materials. For the purpose of lowering the self-citation rate, authors ought to cite published material more frequently.

# Response: Thank you for highlighting these areas for improvement, to meet the standards of scientific rigor and integrity. We appreciate the thorough evaluation of the introduction. Upon careful consideration of your feedback, we improved the logical structured of the introduction that effectively justifies the selection of SETD2 as the study subject. We also revise the introduction to better elucidate the rationale behind choosing SETD2 and establish a clear connection between SETD2, genetic aberrations, and biomarkers. Additionally, we acknowledge the importance of diversifying our references and reducing the self-citation rate. We need to highlight that our lab working with prostate cancer for more than a decade and we are one of few labs dedicating more research in translation of PCa and those articles been cited is relevant to the outcomes. However, in the revised manuscript, we will incorporate additional relevant literature to support our findings and lower the percentage of self-citations.

3. The authors did not go into detail regarding SETD2 protein expression scoring in Materials and Methods Section 2.3. The scoring standards are merely stated. More specifically, how are samples with high expression levels found? Is a field of view considered to have a high-expression sample if it contains a significant number of positively stained cells, or is there another way? Put differently, how do you handle inconsistent expressions in the field of view while scoring? I think that in real testing, this is a common occurrence. For instance, all positive cells exhibit inconsistently high staining signals in Figure 1A, which displays the IHC staining data for GG4_High. Furthermore, the author must include trustworthy sampling information, such as the number of sections that were obtained for each sample's IHC staining, the number of fields of view that were examined, etc. It is crucial to consider the number of slices and viewing windows. Because the relative density score of the SETD2 expression is essentially around 2, as result 3.1 shows. The sample size affects the statistical significance between the comparison groups in Figures 1B and 1C.

# Response: The TMA were constructed using two replicate 0.6 mm cores from the two most common GG areas per each patient’s tumor. Evaluation of IHC categorical intensity was based on the predominant intensity observed across each core and then averaged per total cores for each patient.

4. Consequently, Part 3.6's conclusion is incredibly weak. Figure 4A indicates that there are variations in survival between various clinical groups, based on the legend. However, it is impossible to determine whether the variations in survival across various clinical groups are statistically different because significance testing is not used. Furthermore, there is no connection between the expression of SETD2 and Figure 4A. However, Figure 4B does not have significance testing, so it is not possible to conclude that SETD2 expression rises as the Gleason score increases. Additionally, this section only summarizes the findings of several analyses with reference to enrichment analysis. The author jumps to conclusions without bothering to connect the observable findings to particular occurrences. For instance, according to the authors, SETD2 expression is linked to treatment resistance and a dismal prognosis. I'm not sure how such a determination is made.

# Response: Thank you, p values and significance test added.

In addition, a few small recommendations and queries are stated below.

· As may be seen in Figures 1D and 1E, the author combined Scores 1 and 2 into a single group. Why is it done in this manner? It is not stated whether this treatment is necessary. Why not combine points two and three?

 # Response: We appreciate the reviewer's inquiry regarding the grouping of Scores 1 and 2 into a single group and the rationale behind this decision. The decision to combine Scores 1 and 2 into a single group was based on prior analysis where we observe no significant difference between scores 1 and 2 compared to events and due to the fact that there was less separation between those two scores. Additionally, the numbers of cores with score 1 and 2 combined would provide similar number of cases to those observed in group 3 to allow for meaningful analysis based on balanced number of cores between those two groups.

· Markers for the significance test ought to be added to Figures 1B and 1C. Analogously, significance testing is also necessary for Figures 1D and 1E.

# Response: p value was added as per reviewers’ recommendation and also more details about the samples and biostatistics added in the supplementary table 1.

· Does the description in the results section 3.2 correspond to the results in Table 2?

# Response: Yes, the section 3.2 is for the table 2 and we added the proper citation.

· The explanations of the two results are virtually duplicated in the first paragraph of Results 3.3. In essence, Table 3 is Figure 2's data table format.

# Response: Thanks. The table 3 showcased the full details but figure 2 showing the KM curves as visual representation.

· Why is the analysis based on the TCPA data set not using the four-tiered system? Rather, the expression is merely separated into categories based on high and low expression.

# Response: for TCPA data we used Optimal Cutoff(+HR) and grouped as high / low expression

Comments on the Quality of English Language

There are errors in the use of punctuation marks and spelling of words.

# Response: The revised manuscript was edited carefully and proofread for errors, punctuation marks and spelling of words.

Reviewer 4 Report

Comments and Suggestions for Authors

The manuscript exploring the prognostic significance of SET domain containing 2 (SETD2) expression in advanced and castrate resistance prostate cancer discussed the usability of SETD2 as a prognostic marker and its association with other biomarkers such as PTEN, ERG, and p53. It is important to find new biomarkers to identify high-risk prostate cancer cases. The manuscript is well-designed and mostly clearly written. However, I found many places typo mistakes and the most important and frequent error was with SETD2. In many places, it is written as SEDT2.

1.     Another language-related error was in the methodology section 2.1, under the heading Tissue Microarray Construction. The first sentence of the paragraph states, “We analyzed the total of 202 patients diagnosed with prostate cancer, who were treated non-surgically by ADT for disease progression”. Please check this sentence to confirm.

2.     The result, section 3.1, (even the title of this subsection of the result has an error in SETD2), states that no cases of prostate cancer showed an absence of SETD2 expression. It should be written clearly to state that all the specimens showed SETD2 expression with varied expression levels ranging from dim to high.

3.     Please check the title of section 3.5 in the result.

4.     In the discussion section, the last sentence of the paragraph starting from Beyond biomarker discovery….. In the last sentence of the paragraph, Metformin induces SETD2 activity. This sentence also has an error in writing SETD2.

5.     Figure 1A shows GG1 with weak SETD2 and 2 images of GG4 with high. Please check if these are appropriately selected to only show weak and high expressions.

6.     The purpose of analyzing SETD2 in alive vs deceased makes no sense currently. Please clarify what it signifies.

7.     Axes labels on Fig 1B and 1C are missing.

8.     Figure number citation in the last sentence of result section 3.1, about OS and CSS are 1D-E, not 1C-D.

9.     The manuscript states that high expression of SETD2 is associated with overall and cause specific survival. Could authors confirm if they want to state that high expression correlates with favorable or poor OS? It is also not clear what is the expression level in SETD2 high risk and low risk. Please define these groups in terms of the expression level of SETD2. Although, there is information at different places that SETD2 expression is an indicator of poor prognosis.

Comments on the Quality of English Language

Needs correction of many typographical and language-related errors. Some of those are identified in the comments. I highly recommend detailed proofreading.

Author Response

The manuscript exploring the prognostic significance of SET domain containing 2 (SETD2) expression in advanced and castrate resistance prostate cancer discussed the usability of SETD2 as a prognostic marker and its association with other biomarkers such as PTEN, ERG, and p53. It is important to find new biomarkers to identify high-risk prostate cancer cases. The manuscript is well-designed and mostly clearly written. However, I found many places typo mistakes and the most important and frequent error was with SETD2. In many places, it is written as SEDT2.

1. Another language-related error was in the methodology section 2.1, under the heading Tissue Microarray Construction. The first sentence of the paragraph states, “We analyzed the total of 202 patients diagnosed with prostate cancer, who were treated non-surgically by ADT for disease progression”. Please check this sentence to confirm.

# Response: We appreciate the reviewer's feedback and constructive suggestions toward enhance the overall readability and coherence of our manuscript. We extensively edit and refine the revised version of our manuscript. Section 2.1 were re-edited.

2. The result, section 3.1, (even the title of this subsection of the result has an error in SETD2), states that no cases of prostate cancer showed an absence of SETD2 expression. It should be written clearly to state that all the specimens showed SETD2 expression with varied expression levels ranging from dim to high.

# Response: Thank you. Section 3.1 was re-edited.

3. Please check the title of section 3.5 in the result.

# Response: Thanks. corrected

4. In the discussion section, the last sentence of the paragraph starting from Beyond biomarker discovery In the last sentence of the paragraph, Metformin induces SETD2 activity. This sentence also has an error in writing SETD2.

5. Figure 1A shows GG1 with weak SETD2 and 2 images of GG4 with high. Please check if these are appropriately selected to only show weak and high expressions.

# Response: Yes, those are just examples of low and high as the expression of the tree groups (1-3) of SETD2 was observed across all GG and not bound to specific GG being low or high.

6. The purpose of analyzing SETD2 in alive vs deceased makes no sense currently. Please clarify what it signifies.

# Response: Thank you. Since we are looking on prognosis and outcomes, it’s important to mention the SETD2 expression and associated lethality.

7. Axes labels on Fig 1B and 1C are missing.

# Response: Thank you for feedback. Missing labels on Fig 1B and 1C were added.

8. Figure number citation in the last sentence of result section 3.1, about OS and CSS are 1D-E, not 1C-D.

# Response: Thank you. Corrected.

9. The manuscript states that high expression of SETD2 is associated with overall and cause specific survival. Could authors confirm if they want to state that high expression correlates with favorable or poor OS? It is also not clear what is the expression level in SETD2 high risk and low risk. Please define these groups in terms of the expression level of SETD2. Although, there is information at different places that SETD2 expression is an indicator of poor prognosis.

# Response: Thank you. Revised and corrected.

Comments on the Quality of English Language

Needs correction of many typographical and language-related errors. Some of those are identified in the comments. I highly recommend detailed proofreading.

# Response: The English was revised by native speakers, typos and other issues were fixed.

Reviewer 5 Report

Comments and Suggestions for Authors

PSA is the leading biomarker for diagnosis, prognosis and risk classification, a PSA values are required to describe the case series.

Author Response

PSA is the leading biomarker for diagnosis, prognosis and risk classification, a PSA values are required to describe the case series.

# Response: We acknowledge the significance of PSA values in diagnosis and risk classification. However, PSA values is not well-known marker to associate with lethal disease in non-surgical cohorts, or overall survival. Additionally, our study involves non-surgical cohort with incidental and advanced disease detected in clinical scenarios without having PSA levels being assessed prior in clinical decisions, so it is not possible to obtain PSA values in those patients to investigate if pre diagnostic PSA values are associated with lethal disease, which we still not to show significance based on published data.

Round 2

Reviewer 2 Report

Comments and Suggestions for Authors

The writing of the manuscript has been improved. But the description of figures and quality of the figures are still bad.

1.     Some texts needed to be further edited. Such as “Fig 2”.

2.     The quality of Figure 4 requires significant improvement. Specifically, the font size is too small, which hampers the legibility and interpretation of the data presented. Additionally, the figure's descriptions lack clarity. For instance, in Figure 4B, there is no explanation provided for the terms “Clinical_group 6,7,8,9,10”. Furthermore, the labels in Figure 4C, including the Top 50 genes and the title, are indiscernible. These issues need to be addressed to ensure the figure effectively communicates the intended information.

Author Response

The writing of the manuscript has been improved. But the description of figures and quality of the figures are still bad.

Response # Thank you. We have improved the descriptions and figures as well. Figures with high resolution were uploaded separately to improve the  as well.

  1. Some texts needed to be further edited. Such as “Fig 2”.

Response # Thank you. Corrected.

  1. The quality of Figure 4 requires significant improvement. Specifically, the font size is too small, which hampers the legibility and interpretation of the data presented. Additionally, the figure's descriptions lack clarity. For instance, in Figure 4B, there is no explanation provided for the terms “Clinical_group 6,7,8,9,10”. Furthermore, the labels in Figure 4C, including the Top 50 genes and the title, are indiscernible. These issues need to be addressed to ensure the figure effectively communicates the intended information.

Response # Thank you. Figure 4 optimized and revised as recommended by the reviewer. The Clinical group represents Gleason score.

Reviewer 3 Report

Comments and Suggestions for Authors

The author provided direct and precise answers to my questions. I am content with the majority of the responses. Nevertheless, there remain some apprehensions. I recommend implementing alterations in this area to adhere to the specifications for publication.

1. Despite the author's claim of providing high-quality photos, the updated paper still exhibits inadequate image resolution. For instance, the legends of Figure 1D, Figure 1E, and Figure 2. Furthermore, several text annotations in Figure 4 are excessively diminutive, rendering them illegible.

2. Figure 4B still lacks a statistically significant test mark.

3. The self-citation rate has not decreased. The present manuscript includes 7 references, all authored by the same team, specifically references 3, 4, 5, 21, 22, 32, and 36. The rate of self-citation is 19.4%, which is even greater than the rate in the manuscript before it was revised.

Author Response

The author provided direct and precise answers to my questions. I am content with the majority of the responses. Nevertheless, there remain some apprehensions. I recommend implementing alterations in this area to adhere to the specifications for publication.

Response # Thank you.

  1. Despite the author's claim of providing high-quality photos, the updated paper still exhibits inadequate image resolution. For instance, the legends of Figure 1D, Figure 1E, and Figure 2. Furthermore, several text annotations in Figure 4 are excessively diminutive, rendering them illegible.

Response # Thank you. Legends are revised. New figures 4 replaced the old one with larger font size and resolution.

  1. Figure 4B still lacks a statistically significant test mark.

Response # P value added in the revised manuscript.

  1. The self-citation rate has not decreased. The present manuscript includes 7 references, all authored by the same team, specifically references 3, 4, 5, 21, 22, 32, and 36. The rate of self-citation is 19.4%, which is even greater than the rate in the manuscript before it was revised.

Response # Thank you. We removed some of those citations as recommended by the reviewer, however, as early mentioned we only cited relevant articles.  

Reviewer 5 Report

Comments and Suggestions for Authors

Authors:We acknowledge the significance of PSA values in diagnosis and risk classification. However, PSA values is not well-known marker to associate with lethal disease in non-surgical cohorts, or overall survival.

Reviewer : PSA is the reference clinical marker and should be used as reference for introducing other markers.

Authors: Additionally, our study involves non-surgical cohort with incidental and advanced disease detected in clinical scenarios without having PSA levels being assessed prior in clinical decisions, so it is not possible to obtain PSA values in those patients to investigate if pre diagnostic PSA values are associated with lethal disease, which we still not to show significance based on published data.

Reviewer: PSA has to be obtained for all patients suspected for Prostate cancer

Author Response

Authors: We acknowledge the significance of PSA values in diagnosis and risk classification. However, PSA values is not well-known marker to associate with lethal disease in non-surgical cohorts, or overall survival.

Reviewer : PSA is the reference clinical marker and should be used as reference for introducing other markers.

Authors: Additionally, our study involves non-surgical cohort with incidental and advanced disease detected in clinical scenarios without having PSA levels being assessed prior in clinical decisions, so it is not possible to obtain PSA values in those patients to investigate if pre diagnostic PSA values are associated with lethal disease, which we still not to show significance based on published data.

Reviewer: PSA has to be obtained for all patients suspected for Prostate cancer

Response # We appreciate the reviewer comment. However, as this cohort represent real clinical presentation of patients within public health system in Canada, PSA data are not readily available for all patients, as majority of patients with incidental and advanced disease came to clinical attention without suspicious of prostate cancer, hence PSA levels were not assessed. The PSA data is not available for all patients in this cohort and cannot be obtained. Additionally, it is not expected that PSA values will be prognostic in this cohort since PSA values will encompass wide range of different clinical scenario and would be subjected to hormonal changes in some but not all patients. Therefore, biomarkers related to lethal disease will not be accurately assessed to PSA values in this cohort.

We hope this response is satisfactory for the reviewer.